# Impact of rural soundscape on environmental restoration: An empirical study based on the Taohuayuan Scenic Area in Changde, China

**Hui Yang, ShuangQuan Zhang** *

College of Tourism, Central South University of Forestry & Technology, Changsha, China

* T20111197@csuft.edu.cn

**Data Availability Statement:** All relevant data are within the manuscript and its Supporting Information files.

## Abstract

Previous studies on environmental restorative effects have mainly focused on visual landscapes, and less on the influence of soundscapes on restorative, but soundscapes play a crucial role in restorative environments, especially rural soundscapes, but there is insufficient existing theoretical evidence on the subject. Therefore, this study aims to investigate the influence of Rural Soundscape Perception on Environmental Restoration Perception, and introduces two affective variables, tourism nostalgia and place attachment, to explore the mechanism of Rural Soundscape Perception on Environmental Restoration Perception, as well as the moderating role of the number of trips is also discussed. Based on the theory of restorative environment, this study took the Taohuayuan Scenic Spot in Changde, Hunan Province, China, as the case site, and selected the rural soundscape in the area as the research object; a total of 506 valid data were collected through questionnaire surveys, and structural equation modeling was used to validate the collected data. It was found that rural soundscape perception had a significant positive effect on tourism nostalgia, place attachment, and environmental restoration perception. The results also showed that tourism nostalgia and place attachment mediated the relationship between rural soundscape perception and environmental restoration perception. Additionally, the results revealed that the number of trips did not play a moderating role in the structural relationship between rural soundscape perception and environmental restoration perception. Last, the results of the study shed light on the complex influence path of "rural soundscape perception→tourism nostalgia→place attachment→environmental restoration perception", which provides a new perspective for understanding the mechanism of the rural environment to people's health, and also has a certain guiding significance for the landscape planning of rural tourism sites.

## 1. Introduction

With the rapid urbanization process, various environmental problems are becoming more and more prominent, among which, the noise problem is getting more and more attention. Studies have shown that the hustle and bustle of the modern urban living environment not only affects the psychological state and emotional health of urban residents, making people inwardly

**Funding:** This study was funded by Hunan Natural Science Foundation (No. 2023JJ31017), Special Funding for Basic Education Development from Hunan Provincial Department of Finance (2022No.69), and Key Projects of Hunan Provincial Department of Education (No. 23A0232).

**Competing interests:** The authors have declared that no competing interests exist.

irritable, breeding negative emotions and difficulty in concentrating, but also leads to the spread of chronic diseases such as insomnia, obesity, depression, etc. [1]. At the same time, the high-density development of urban space leads to a decrease in the natural environment, urban people have fewer opportunities to contact nature, and the mental pressure brought by the fast-paced urban life is increasing [2]. Under such psychological conditions, people see rural tourism as a carrier to return to nature and yearn for the countryside and nature [3]. Different from the urban environment filled with mechanical sounds, the countryside is filled with a large number of natural sounds such as running water, wind, birdsong, insects and other natural sounds, as well as the humanistic sounds generated by farmers' life and production, etc. These are highly rated sounds in the outdoor environment [4, 5], which can enable people to feel the beauty of nature more holographically, and this is a very important factor for tourists to look forward to the countryside and love the countryside tourism.

At present, rural tourism has become the main way of tourism for Chinese residents, and the rural sound environment is gradually emphasized by tourists [6–8]. The concept of "Soundscape" was first proposed by the Finnish geographer Granoe in 1929, which is used to describe the "listener-centered sound environment" [9]. At the end of 1960s, the Canadian musicologist Schafer defined the soundscape as a soundscape of the countryside. Schafer defined soundscape as "a sound environment that is perceived and understood by human or social participation" [10]. Early tourism scholars due to the "tyranny of the visual" and lead to the soundscape and other sensory aspects of the research is not yet in place [11], the research on Rural Soundscape Perception is mostly concerned about the design and improvement of the visual landscape, ignoring the construction of auditory Soundscape Perception and its implied value of the analysis [12], the visual perception of soundscape Perception. the visual perception factor is certainly important, but the sound landscape represented by auditory perception should not be ignored. Sound is a key link in enriching and enriching tourists' local perception and experience [13], which not only affects tourists' experience through physical properties such as sound level, but also has cultural and social existence [14]. It has been confirmed that the purpose of rural tourism for tourists is to enjoy the peaceful rural soundscape and return to rural idyllic life [14]. Liu (2020) [15] found that the rural sound environment can not only provide a comfortable physiological experience, but also form the emotional resonance of "nostalgic memory". Wang (2023) [16] suggests that rural soundscape is an important part of rural tourism for tourists, and how to optimize and create a satisfactory rural soundscape for tourists has become an urgent problem for scholars. Thus, rural soundscape is very important in rural tourism destinations. At present, the research on rural soundscape mainly focuses on Rural Soundscape Evaluation [17–19], Sound Preference [20,21], Soundscape Protection and Design [22–24], Rural Soundscape Perception [25,26], and there are relatively few researches with the theme of Rural Soundscape Restorative Effect. However, the rich sound resources in the countryside can not only create a good tour experience for tourists, but also have a role in promoting the health of urban residents that cannot be ignored. Therefore, in view of the fact that there are fewer studies on soundscapes in the field of rural tourism and the limited number of studies discussing the restorative effects of rural soundscapes in the context of rural tourism, the present study argues that it is necessary to pay more attention to the soundscape of rural tourism destinations and the combination of rural soundscapes and environmental restorative effects. "Soundscape" exists through people's perception of the sound environment of a place [27]. Based on previous research, this study further rationalizes and integrates the concept of "Rural Soundscape Perception", which is defined as the perception and understanding of the surrounding sound environment by tourists during rural tourism.

Since the 20th century, restorative environments have gradually become a research hotspot in many fields, and have received more and more attention from the fields of environmental

psychology, public health, and urban planning, etc. [28,29]. In 1983, Kaplan et al. proposed the concept of restorative environments for the first time, which means that restorative environments are those that can help people alleviate mental stress, reduce negative emotions, and physical and mental exhaustion. Kaplan et al. [30] first proposed the concept of "restorative environment" in 1983, that is, an environment that can help people relieve mental stress, reduce negative emotions and physical and mental fatigue, and then empirical studies on the restorative effect of the environment have been emphasized, and a large number of studies have confirmed that natural environments are effective in relieving stress and recovering from fatigue [31].

Environmental restorative effects refer to the positive outcomes that may occur when a person is in a natural environment characterized by recovery-promoting, stress-reducing, and restorative features [32]. This feeling that a person in a particular environment will produce an Environmental Restoration Perception that results in positive emotions such as relief of mental stress, reduction of negative emotions and physical and mental exhaustion is known as Environmental Restoration Perception [33]. The subject of Environmental Restoration Perception is a person and the object is the environment with restorative effects [34]. The relevant research results about Environmental Restoration Perception have become increasingly rich in recent years, and the research mainly includes the influence and action mechanisms of restoration, the quantification of environmental restoration, and the interaction relationship between environmental preference and restoration [35–38]. Researchers have focused on the restorability of individuals in different types of environments, such as natural environments (parks, villages, green spaces, etc.) [39], urban environments (museums, zoos, shopping centers, etc.) [40], and restorability measures in specific environments [41]. There are also studies that focus on the influences on Environmental Restoration Perception and the results of their effects. For example, place memory, place dependence, and place identity have all been shown to positively influence people's Environmental Restoration Perception [42]. In addition, familiarity with the environment and length of stay in a restorative environment both influence an individual's Environmental Restoration Perception [43]. In terms of sensory stimulation, early studies of Environmental Restoration Perception theory have focused on the level of visual perception of restoration [45]. Viewing natural landscapes has been shown to contribute to improved mood, attentional recovery and more effective recovery from physiological stress and health [44,45]. Students who learn in environments where they can view natural scenery may improve their concentration [46] or perform better in school [47]. However, the role of soundscapes in enhancing concentration, relieving stress, and improving mood should not be overlooked as well. Compared to visual scenes, in some cases sound has a more significant effect on people's recovery from fatigue and worry [48]. With the rise of the soundscape research field, more and more scholars have begun to pay attention to the restorative effects of auditory stimuli, and have gradually confirmed the health restorative effects of soundscape in various aspects such as physiology and psychology [49,50], and concluded that soundscape, as an important environmental element, can have a significant impact on the restorative quality of the environment [51]. For example, Li et al. (2019) [52] found that natural sounds decreased heart rate, respiratory rate, and respiratory depth, and increased R-wave amplitude, heart rate variability, and brain wave α and β values; Zhao Guard et al. (2019) [53] confirmed in a study of audiovisual interactions that increasing birdsong in parks with flat topography is an effective way to improve spiritual healing; Hu et al. (2021) [54] studied urban open space after studying the restorative effects of soundscapes in urban open spaces, suggesting that soundscapes play a crucial role in restorative environments. The countryside is an important type of restorative environment [55], with characteristic soundscapes such as the sound of running water, wind, insects and birds [56]. It has been shown that rural soundscapes have the highest restorative

potential among scenarios of urban soundscapes, urban park soundscapes, and rural soundscapes [57]. Watts et al. (2013) [58] investigated that the countryside is a restorative or serene environment, which can alleviate cognitive overload and reduce stress. Sang et al. (2020) [59] experimentally confirmed, through virtual reality, that rural soundscapes are able to exert a restorative effect on human psychological and physiological restorative effects, which shows that the restorative effects of rural soundscape are of great research value. However, previous studies have mainly focused on the restorative effects of soundscapes in urban spaces [60–63], such as city parks, forest parks, community parks and campuses, and fewer scholars have studied the restorative effects of rural soundscapes on the environment, while rural tourist destinations, as an important place for city dwellers to alleviate their stress and relax, assume the role of providing restorative environments, and their restorative research is justified. As an important place for urban residents to relieve stress and relax, rural tourist destinations assume the role of providing a restorative environment, and their restorative research deserves attention.

Although a certain amount of research has been carried out on the restorative effect of rural soundscape in the past, the mechanism of rural soundscape's influence on the restorative effect on the environment is still unclear [64], and the guiding significance for rural landscape planning is limited. Therefore, it is necessary to deeply study the influence mechanism of rural soundscape on environmental restorative effect. This study notes that the environmental restorative effect of rural soundscape will be influenced by emotional factors, such as tourism nostalgia and place attachment. This is because the connection between people and the environment is not only highly related to the physical characteristics of the environment, but also inseparable from the people and the subject's own experience in that environment, which is the result of the interaction between the environment and the cognitive, emotional, behavioral, and social components of people [65]. Nostalgia is a word derived from two Greek roots "Nostos" and "Algos", the former meaning home, homecoming, and the latter referring to an agonizing state of being anxious to return home. Tourism is an important way of generating nostalgia, and a happy emotion accompanied by sadness that people experience during tourism related to past experiences with the place or the tourists themselves is called tourism nostalgia [66], while place attachment refers to the degree to which a person maintains an emotional connection with a spatial area [67]. Therefore, both tourism nostalgia and place attachment are emotional factors that can create a connection between people and their environment. The rural soundscape is an important factor in the generation of nostalgia [12], and some scholars have pointed out that the essence of "nostalgia" is nostalgia, which is people's nostalgia for the local people or their memories of the past [68]. At the same time, "nostalgia" is a typical place attachment, which is a kind of emotional embodiment of tourists' hometown [69]. Thus, it can be seen that rural soundscapes are closely related to nostalgia and place attachment. In addition, Korpela et al. (2001) [70] found that individuals can fully relax and show good recovery function in places with high attachment level, while attention is not easy to recover in places with low attachment. Nostalgia is considered to have restorative functions due to positive factors such as satisfying people's need to belong, enhancing positive emotions, boosting self-esteem, and strengthening social ties [71]. Therefore, nostalgia and place attachment can be generated by rural soundscapes, and nostalgia and place attachment can affect environmental restorativeness, which is closely related to environmental restorativeness. Then, what is the mechanism of rural tourists' Rural Soundscape Perception, tourism nostalgia and place attachment on Environmental Restoration Perception? It has not been explored in depth by the academic community yet.

To summarize, soundscape is very important, but the related research has yet to be in-depth. From the perspective of research, most of the previous environmental restorative studies only emphasize the restorative effects brought by visual landscapes or explore their

restorative effects from the perspective of the overall environment, and fewer studies have examined the effects of individual soundscapes on the human body. From the perspective of research area, most of the soundscape restorative research now is in public spaces such as urban parks, forest parks, college campuses, etc., and there are fewer researches involving rural tourist destinations. In addition, scholars have already confirmed that soundscape has restorative functions [72–74], but there is a lack of research on its influence mechanism with environmental restorative effects. Therefore, in this study, the Taohuayuan Scenic Area in Changde City, Hunan Province was selected as a case study for empirical analysis, with the main purpose of exploring the influence mechanism of Rural Soundscape Perception on Environmental Restoration Perception by constructing a structural equation model, and introducing the two affective variables of tourism nostalgia and place attachment, to provide new research perspectives for the theoretical study of Rural Soundscape Perception, and at the same time, to provide theoretical basis for the design and creation of soundscape in rural tourism destinations and decision-making support. In addition, the reasons for the restorative effect of rural soundscape are still unclear in the academic world, and the introduction of the relationship between the two variables into the rural sound environment for restorative empirical research is a novel and meaningful research topic in itself. This study attempts to determine the impact of Rural Soundscape Perception on Environmental Restoration Perception by answering three research questions: (1) whether Rural Soundscape Perception has an impact on Environmental Restoration Perception; (2) what are the pathways and mechanisms of action of Rural Soundscape Perception on Environmental Restoration Perception; and (3) whether tourism nostalgia and place attachment mediate the effect.

## 2. Development and justification of hypotheses

### 2.1. Rural soundscape perception and environmental restoration perception

The environment can affect an individual's psychological state [75], and environments that enable a person to better recover from negative emotions linked to mental fatigue and stress are known as restorative environments, while experiences in restorative environments are referred to as environmental restorative perceptions [33]. Tourists are exposed to sensory stimuli from their environment during their activities, and soundscape is the most important sensory stimulus other than the visual senses due to its pervasive nature and compulsion to be perceived [76]. It has been demonstrated that appropriate soundscapes are beneficial for people to recover from stress [77], and in particular, the sounds of nature are effective in reducing tension and anxiety [78] as well as the sensation of pain [79]. Payne found that individually perceived soundscapes can provide a restorative experience in urban parks after conducting a questionnaire survey with 400 park visitors [80], and then conducted a study of three scenarios, urban parks and rural areas, to determine how soundscapes can provide restorative experiences in urban parks and rural areas, urban parks and rural areas, and found that rural soundscapes were the most restorative [57]. Ojala et al. (2019) [81] found that for noise-sensitive people, noisy city centers had a negative restorative effect, areas such as urban parks with a mixture of natural and noisy sounds were generally restorative, and areas such as the countryside, with lower sound pressure, had the best restorative effect. natural environments had the best restorative effect; Sang et al. (2020) [59] confirmed that rural soundscapes can have a restorative effect on human psychology and physiology. These studies provide evidence for the positive impact of rural soundscapes on both physical and mental health. Based on the review of the literature, this research proposes the following hypothesis:

**H1:** *Rural soundscape perception has a significant positive effect on environment restoration perception.*

## 2.2. Rural soundscape perception, tourism nostalgia and environmental restoration perception

The basic view of modern cognitive psychology is that human beings are transmitters of information and systems of information processing. One can categorize things into auditory (sound code), visual (shape code), and semantic (perceptual code) based on different characteristics, and it is believed that auditory (sound code) can be used to retrieve previously formed memories through sound [82]. Hall et al. (2013) [85] found that characteristic sounds rich in historical, cultural, and regional geographic attributes evoke individual subjective experiences, past historical memories, and other socio-cultural factors for those who know and are familiar with these sounds. Therefore, sounds not only carry people's memories, but also help people to reproduce their memories, and nostalgia is closely related to reminiscence, which is the reproduction of past life and memories in a positive way [83]. Therefore, those good memories awakened by sound tend to trigger people's nostalgia. It has been shown that soundscape affects human nostalgia [84], and the sound stimulation received by the senses causes people to feel nostalgic [85]. In the process of rural tourism, attractive soundscape has a positive promotion effect on tourism nostalgia [86]. Thus, it can be concluded that there is a positive role relationship between rural soundscape perception and nostalgia.

Nostalgia has been conceptualized in the theoretical framework of modern psychology as a positive emotion with pragmatic functions [87], and it is increasingly becoming an effective way to improve negative perceptions and enhance well-being [88]. It can stimulate positive emotional experiences, enhance positive evaluations, strengthen social ties, foster a sense of belonging, and promote positive emotions like well-being. Ultimately, nostalgia can effectively regulate the high-pressure state of the public and alleviate social anxiety [89,90]. Nostalgia is increasingly becoming an effective way to improve negative perceptions and enhance well-being [88]. Korpela et al. (2008) [91] refer to the phenomenon in which an individual's stress and exhaustion are relieved, positive emotions are increased, negative emotions are decreased, and attention is restored as psychological restoration. Individuals in restorative environments develop environmental restoration perception. Therefore, nostalgia, as a positive emotion, plays a significant role in generating environmental restoration perception. It has been confirmed that nostalgia has a restorative effect [92]; Cao et al. (2023) [88] used a scale to confirm the restorative effect of nostalgia at the psychological level, which can positively and significantly affect tourists' subjective well-being. In addition, Smalley (2022) [93] found that those memories triggered by sounds are psychologically restorative, and if people like these memories, then these memories may bring therapeutic effects, and nostalgia is closely related to memories. Therefore, it can be assumed that there is a positive role relationship between nostalgia and Environmental Restoration Perception.

The mediating role of nostalgia is now supported by most studies [94–96]. As mentioned above, soundscape triggers nostalgia, which in turn promotes Environmental Restoration Perception. In addition, the nostalgia that people experience during tourism is referred to as tourism nostalgia [97], i.e., tourism nostalgia still essentially belongs to a nostalgic emotion, set only in the particular context of tourism. Therefore, this study considers tourism nostalgia as a key variable of Rural Soundscape Perception acting on Environmental Restoration Perception and verifies its mediating effect, so the following hypothesis is proposed: the nostalgia that people experience during tourism is referred to as tourism nostalgia [97].

**H2:** *Rural soundscape perception has a significant positive effect on tourism nostalgia.*

H2a: *Rural soundscape perception has a significant positive effect on personal nostalgia.*

H2b: *Rural soundscape perception has a significant positive effect on historical nostalgia.*

**H3:** *Tourism nostalgia has a significant positive effect on environmental restoration perception.*

H3a: *Personal nostalgia has a significant positive effect on environmental restoration perception.*

H3b: *historical nostalgia has a significant positive effect on environmental restoration perception.*

**H4:** *Tourism nostalgia mediates between rural soundscape perception and environmental restoration perception.*

## 2.3. Rural soundscape perception, place attachment and environmental restoration perception

Soundscape can directly produce emotional experience [12]. The theory of place holds that place is a space with special meaning for tourists, and when tourists produce a good tourism experience in this place, they will produce psychological connections such as a sense of dependence and identity [98], and perception also serves an important role in the process of fostering feelings of attachment [99]. Then, travelers may develop place attachment emotions after getting a good Rural Soundscape Perception experience in a rural tourist place. It has been confirmed that soundscape creates a sense of cultural identity and attachment to a place [100]. Bartos et al. (2013) [101] used ethnographic methods to investigate and found that children and adolescents formed place attachment feelings under the influence of soundscape; Zhang Jie et al. (2018) [102] concluded by analyzing the impact of soundscape on tourism social psychology that soundscape perception enhances tourism satisfaction and also strengthens travelers' place attachment, allowing them to identify with the destination; Fang Shumiao et al. (2022) [103] found that rural sound as a perceived value of rural tourism positively affects place attachment. In addition, Xu Hong et al. (2020) [104] found that the perception of odor landscape in rural tourism helps tourists to generate place attachment emotion, and the tourist's tourism process is always carried out in a certain sound environment, sound symbols, together with other sensory symbols such as visual symbols and olfactory symbols, constitute the complete experience of the tourist [76], so the sound landscape also has a certain place attachment emotion influence. Based on the above, this research concludes that the perception of rural soundscapes promotes place attachment.

On the other hand, Korpela et al. (1996) [105] found that places that individuals preferred tended to have stronger positive affective associations with them. Subsequently, Korpela et al. (2001) [70] further demonstrated that environmental preferences have a strong influence on environmental restoration perception, with individuals tending to have higher environmental restoration perception of their favorite places than of places they dislike. Therefore, place attachment may be associated with environmental restoration perception. It has been shown that the place identity and place dependence dimensions of place attachment are positively related to environmental restoration perception [106]. Liu et al. (2019) [107,108] found that urban residents' place attachment to recreational parks positively affected their environmental restoration perception. Xi et al. (2021) [109] took vacation tourists as the research object and showed that place attachment positively affected environmental restoration perception. Therefore, it can be hypothesized that place attachment also promotes environmental restorative perceptions.

The mediating role of place attachment is a key issue in academic research [42]. As mentioned above, rural soundscape perception promotes place attachment, and place attachment promotes environmental restoration perception, so rural soundscape perception may ultimately stimulate environmental restoration perception through the mediating bridge role of place attachment. Based on this, this study proposes the following hypotheses:

**H5:** *Rural soundscape perception has a significant positive effect on place attachment.*

H5a: *Rural soundscape perception has a significant positive effect on place dependence.*

H5b: *Rural soundscape perception has a significant positive effect on place identity.*

**H6:** *Place attachment has a significant positive effect on environmental restoration perception.*

H6a: *Place dependence has a significant positive effect on environmental restoration perception.*

H6b: *Place identity has a significant positive effect on environmental restoration perception.*

**H7:** *Place attachment mediates between rural soundscape perception and environmental restoration perception.*

## 2.4. Chained multiple mediation of tourism nostalgia and place attachment

Nostalgia is a happy emotion related to the past and accompanied by sadness that people commonly experience in their daily lives [66], and place attachment is mainly an emotional connection that people have to places [110]. Therefore, tourism nostalgia and place attachment are fundamentally emotional states that are psychological variables in the influence mechanism. Some scholars believe that the psychological factors affecting tourists contain various complex variables, and considering the impact of only one variable may not be comprehensive enough. Even if considering multiple intermediate variables with an independent intermediary role is not entirely consistent with the actual situation, one should still consider multiple intermediary variables with multiple intermediary effects and intermediary variables in between for the chain effect [103]. Tang et al. (2020) [111] found that the three mediating variables of learning gain, tourist perceived value, and tourist satisfaction played chained multiple mediating roles between the quality of cultural heritage revitalization experience and destination loyalty after researching scenic areas such as Jinggang Mountains. Long et al. (2021) [112] investigated and found that nature connectedness and restorative perception had chained mediating roles in the effects of perceived environmental aesthetic quality on positive emotions. The above scholars' studies confirmed the necessity of chained multiple mediating roles. Therefore, this research also included two psychological variables, tourism nostalgia, and place attachment, in the discussion of chained mediating roles.

The previous hypothesis analysis shows that tourism nostalgia and place attachment each produce separate mediating effects. However, these two mediating variables may also affect each other, i.e., there is an ordered chain of mediators. Yeh et al. (2012) [113] argued that items associated with history can impart historical knowledge to tourists and provide them with an experience of a particular historical era. This can then trigger nostalgia and stimulate emotions related to place attachment. In their study, Gao et al. (2017) [114] demonstrated a significant and positive effect of nostalgia emotion on place attachment among intellectually-oriented youth. Zhu et al. (2019) [115] confirmed that the nostalgic emotions generated by tourists visiting ancient villages during tourism have a significant impact on the tourists' attachment to the place and their place identity. These studies confirm that tourism nostalgia has a positive effect on place attachment. Based on this, the research puts forth the following hypotheses:

**H8:** *Tourism nostalgia has a significant positive effect on place attachment*

**H9:** *Chain-mediated effects of tourism nostalgia and place attachment between rural soundscape perception and environmental restoration perception*

## 2.5. Moderating effect of the number of trips

Chen et al. (2006) [116] concluded that subjective factors such as the number of trips and past experiences affect tourists' recreational experience and satisfaction. Huang (2008) [117] used Vietnam as a case study to confirm that the more times traveled in Vietnam, the higher the satisfaction of tourists. Huang (2015) [118] found through his study that positive emotions such as interest involvement in tourism, tourism satisfaction, and subjective well-being increase accordingly as tourists travel more often. Furthermore, previous travel experiences have been found to alter repeat travelers' perception of the destination image [119] and impact their attitudes and behaviors towards the destination [120,121]. These studies affirm the significant impact of the number of trips on tourism experience, tourist satisfaction, and positive emotions. Tourists always travel in a particular sound environment, and numerous studies have verified the restorative benefits of soundscapes. Therefore, is rural soundscape perception affected by the number of tourist trips, which in turn affects environmental restoration perception?

**H10:** *The number of trips has a significant positive moderating effect on the relationship between rural soundscape perception and environmental restoration perception.*

Based on the above research hypotheses, a model of causal relationships was constructed, as shown in Fig 1.

## 3. Materials and methods

### 3.1. Overview of case sites

Taohuayuan Scenic Area (Taohuayuan), also known as "the Peach Blossom Land", is situated in the southwestern part of Taoyuan County, Changde, Hunan Province, China. Taohuayuan

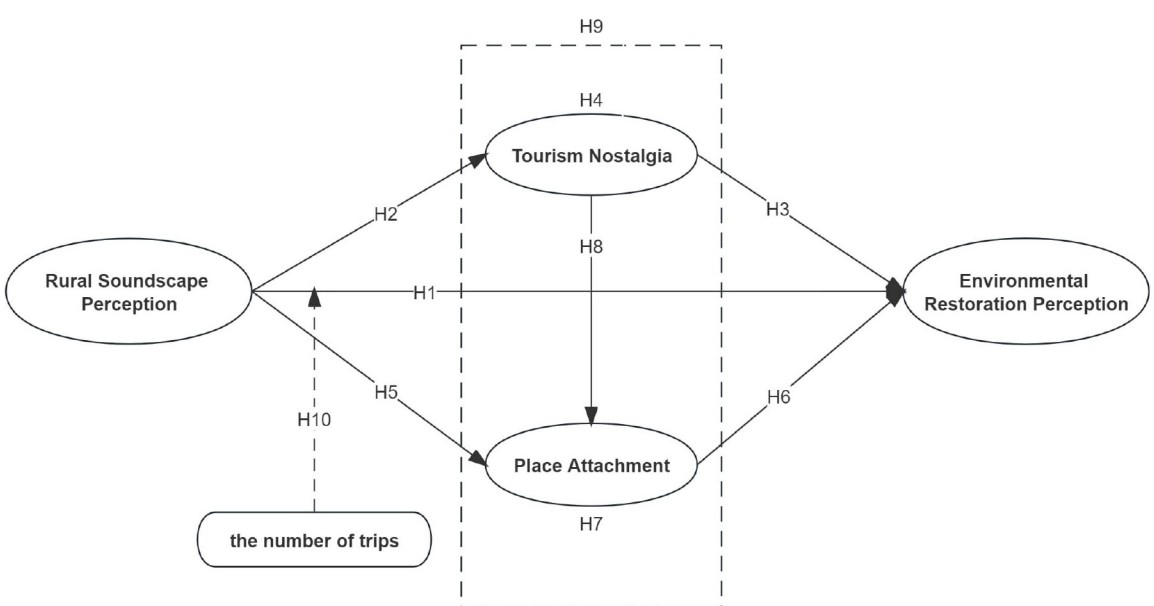

**Fig 1. Research model.**

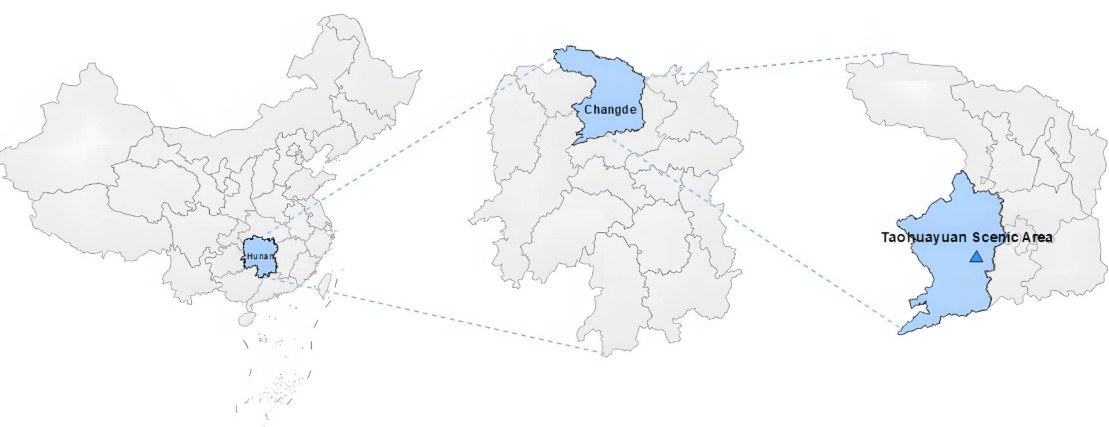

**Fig 2. Location of Taohuayuan scenic area.**

is located at the latitude of 28˚47′-28˚49′ north and the longitude of 110˚25′-110˚27′ east, with a total area of about 157 square kilometers, and is surrounded by many scenic areas, such as Zhangjiajie, Yuelu Mountain, and Hengshan Mountain, to name a few, The relevant information is shown in Fig 2. Taohuayun originated from the famous poet Tao Yuanming's "*The Peach Blossom Spring* in the Eastern Jin Dynasty of China, which describes a worldly paradise and a prototype of idyllic life and is known as "The Peach Spring Beyond this World″ all over the world, meaning a place that is extremely beautiful and where everything seems perfect, especially a place far away from modern life. Taohuayuan has a very deep cultural heritage and is a famous rural tourist place. Huang et al. (2011) [122] believe that the Taoyuan mood is the archetype of rural tourism. Taohuayuan has been honored with the titles of China's National Scenic Area, China's Key Cultural Relics Protection Unit, and China's 5A Grade Tourist Attractions, once awarded China's Top Ten Cultural Scenic Areas at the same time as the Forbidden City and the Potala Palace. Moreover, Taohuayuan has both natural soundscapes and humanistic soundscapes, including the sounds of running water and wind and rain, animal chirping (such as insects and birds), plant branches and leaves, as well as customary performances (such as flower-drum operas, Wuling operas, folk dances, etc.) [123], which constitutes a relatively complete system of Rural soundscapes and is of typical representativeness to the study of Rural soundscapes. Fig 3 shows a photo of a live shot of the peach blossom.

## 3.2. Measurement items

The questionnaire includes five main parts: socio-demographic variables, rural soundscape perception scale, tourism nostalgia scale, place attachment scale, and environmental restoration perception scale. The Tourism Nostalgia Scale draws on the mature scale designed by Chris et al. [124,125] and Jing Xue [126], which is divided into two dimensions, personal nostalgia and historical nostalgia, with a total of 13 items, and the empirical results show that the questionnaire has a good reliability and validity, which is in line with the requirements of psychometrics, and it is a better tool for measuring tourism nostalgia. The place attachment scale is based on the mature scale compiled by Williams et al. (2003), and refers to the research of Xi Wang et al. (2021) [131], combining with the actual situation of sampling, and finally forms the place attachment scale, which is divided into two dimensions, place identity and place dependence, with a total of 9 question items. The Environmental Restoration Perception Scale mainly referred to the revised Revised Perceived Restorativeness Scale (RPRS) of Huang Zhangzhan et al. (2008) [127] and followed the four-dimensional structure of its delineation,

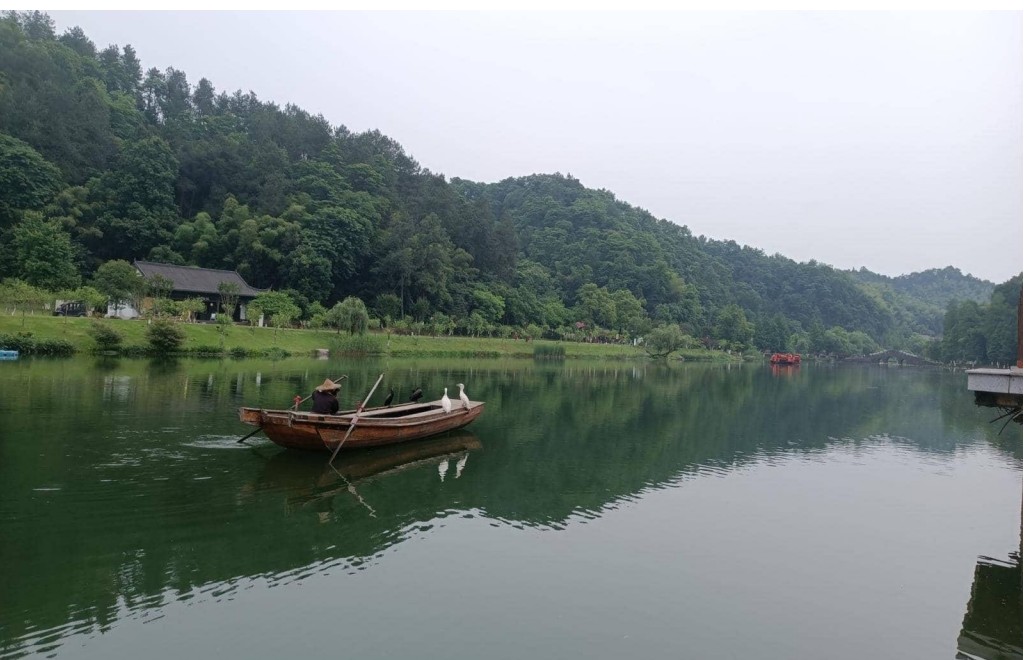

**Fig 3. Taohuayuan scenic area (author's own photo).**

i.e., "charisma, extensibility, compatibility, and distance" The RPRS scale was originally developed by Hartig et al. [128], but due to the semantic ambiguity of six test items in the RPRS scale, Huang et al. revised the RPRS scale, which is more semantically clear and more applicable to the Chinese context. For the Rural Soundscape Perception Scale, since there are fewer scales related to measuring tourists' soundscape perception by scholars, the Rural Soundscape Perception Scale mainly draws on the scale development specifications proposed by Churchill [129], Rossiter [130], etc. as well as Qiu Mengluo et al.'s (2017) [131] soundscape perception scale, and at the same time combines with the conceptual characteristics of Rural Soundscape Perception, developed some measurement items on their own to make up for the shortcomings of the literature. The preliminary construction of the Rural Soundscape Perception Scale consisted of five items, and an online questionnaire pre-survey was conducted, and after excluding invalid questionnaires, a total of 255 valid questionnaires were retrieved. Subsequently, the reliability test of Rural Soundscape Perception Scale was conducted with the questionnaire data. The test result was KMO>0.8, and the significance probability of Butler's spherical test was 0.00<0.01, indicating that the scale was suitable for factor analysis; the validity test showed that the factor loadings of the five indicators were greater than 0.5, which fulfilled the criteria without the need to delete the indicators, and five question items were finally obtained.

The measurement scales were all Likert 5-point scales, with "1" representing strongly disagree, "2" representing disagree, "3" representing average, and "4" for agree, and "5" for strongly agree. Details of the scales are shown in Table 1.

## 3.3. Data collection

The data was collected during a field survey conducted in Taohuayuan between May 19th and 24th, 2023. The team conducted a convenience sample survey on tourists in areas where they were relatively concentrated and where the rural scenery was typical. A total of 530

**Table 1. Measurement items.**

| Variables | Variable quantity | Potential items | References |
|---|---|---|---|
| Rural soundscape perception | RSP1 | I can feel the country life in the sound | Churchill [129] Rossiter [130] Qiu et al. [131] |
| | RSP2 | I can hear the sounds of the countryside clearly | |
| | RSP3 | I hear certain sounds and immediately think of certain country scenes | |
| | RSP4 | I can feel the nature, history and culture of the countryside in the sound | |
| | RSP5 | I can tell which sounds are pleasant or unpleasant | |
| Personal nostalgia | PN1 | Rural sounds remind me of happy experiences from my formative years | Chris et al. [124,125] Xue [126] |
| | PN2 | Rural sounds bring back memories of my formative years | |
| | PN3 | Rural sounds remind me of friends from the past | |
| | PN4 | Rural sounds remind me of past times with my family | |
| | PN5 | Rural sounds evoke feelings I've had before | |
| | PN6 | Rural sounds remind me of the old days | |
| | PN7 | Rural sounds remind me of times gone by | |
| Historical nostalgia | HN1 | Rural sounds remind me of specific historical eras | |
| | HN2 | Rural sounds remind me of scenes of people's lives in the past | |
| | HN3 | Rural sounds make me nostalgic for the traditional customs of the past | |
| | HN4 | Rural sounds make me imagine the lives of previous generations | |
| | HN5 | Rural sounds remind me of things that happened in the distant past | |
| | HN6 | Rural sounds remind me of times long before I was born | |
| Place dependence | PD1 | The countryside is perfect for traveling | Williams et al. [132] Xi et al. [133] |
| | PD2 | The countryside makes me stay longer | |
| | PD3 | I had the best experience in the countryside out of all the places I've visited | |
| | PD4 | I am most satisfied with it of all the places I have visited | |
| | PD5 | I enjoy the countryside more than other places | |
| Place identity | PI1 | I strongly identify with the countryside | |
| | PI2 | Visiting the countryside makes me feel better about myself | |
| | PI3 | The countryside keeps me coming back | |
| | PI4 | The countryside means a lot to me | |
| charisma | CH1 | Rural sounds have appealing qualities | Hartig et al. [128] Huang et al. [127] |
| | CH2 | Rural sounds draw me to more exploration and discovery | |
| | CH3 | Rural sounds are fascinating | |
| | CH4 | I want to spend more time in the countryside because of the rural sounds | |
| ductility | DU1 | Everything I hear in the countryside is harmonious and unified | |
| | DU2 | I am intrigued by rural sounds that I have never heard before | |
| | DU3 | Rural sounds match the environment. | |
| | DU4 | Rural sounds can make me think of many wonderful associations | |
| compatibility | CO1 | I can do what I like to do in the countryside | |
| | CO2 | I can get used to the rural sounds very quickly | |
| | CO3 | I feel like I have become one with the countryside | |
| | CO4 | I can find ways to enjoy myself in the countryside | |
| | CO5 | The countryside is a good place to do the things I like to do | |
| distance | DI1 | The countryside can give me an experience of detachment | |
| | DI2 | The countryside gives me a break from the routine of everyday life | |
| | DI3 | The countryside is a place where I can take a complete break | |
| | DI4 | The countryside is a place where I can relax and unwind | |
| | DI5 | In the countryside, I feel free from the constraints of work and daily life | |

**Table 2. Descriptive statistical results.**

| Indicator | Item | Frequency | % |
|---|---|---|---|
| Household identification | Urban residents | 251 | 49.6 |
| | rural residents | 255 | 50.4 |
| Number of rural tourism experiences, | 1 time | 83 | 16.4 |
| | 2 times | 70 | 13.8 |
| | 3 times and above | 353 | 69.8 |
| Gender | Male | 211 | 41.7 |
| | Female | 295 | 58.3 |
| Age | Below 18 years old | 31 | 6.1 |
| | 18 to 29 years old | 258 | 51.0 |
| | 30 to 39 years old | 95 | 18.8 |
| | 40 to 49 years old | 67 | 13.2 |
| | 50 to 59 years old | 40 | 7.9 |
| | 60 and above | 15 | 3.0 |
| Education | Junior high school and below | 66 | 13.0 |
| | High school or junior college | 88 | 17.4 |
| | Undergraduate or college | 334 | 66.0 |
| | Graduate student and above | 18 | 3.6 |
| Job | National civil servants | 24 | 4.7 |
| | Enterprise and public utility managers | 31 | 6.1 |
| | Unit staff/workers | 71 | 14.0 |
| | Private Owners | 51 | 10.1 |
| | Military personnel | 2 | .4 |
| | Unemployed/Laid-off | 3 | .6 |
| | Students | 216 | 42.7 |
| | Retirees | 9 | 1.8 |
| | Others | 51 | 10.1 |
| | Professionals and technicians (e.g. doctors, teachers) | 32 | 6.3 |
| | Farmers | 16 | 3.2 |
| | National civil servants | 24 | 4.7 |
| Individual monthly income | 1500 and below | 160 | 31.6 |
| | 1501–3000 | 100 | 19.8 |
| | 3001–5000 | 97 | 19.2 |
| | 5001–8000 | 108 | 21.3 |
| | 8001–15000 | 24 | 4.7 |
| | Above 15000 | 17 | 3.4 |

questionnaires were distributed in this survey, 530 questionnaires were returned, 506 questionnaires were valid, and the recovery rate of valid questionnaires was 95.4%. Informed consent was obtained from all participants for this study. Prior to the questionnaire work, the researcher obtained verbal consent from the participants by clarifying the purpose of the questionnaire to the tourists as well as asking them if they were willing to complete the questionnaire. Demographic distributions of respondents are presented in Table 2. Of the 506 valid questionnaires, 49.6% were urban residents and 50.4% were rural residents, which is roughly the same; 69.8% of the respondents had three or more rural tourism experiences; in terms of gender. 41.7% were male and 58.3% were female; in terms of age, the majority of respondents were between 18–29, accounting for 57.1% of the total respondents. About 66% of the respondents have a bachelor's degree or college degree, with a high level of education. The

occupations of the respondents are concentrated in students and employees, accounting for 54.7% of the total respondents. A total of 68.4% of the respondents had a monthly personal income of RMB 3,000 or more.

## 3.4. Research methods

This research adopts a combination of qualitative and quantitative research methods. Based on theoretical research and hypothetical modeling, structural equation modeling was introduced to test the hypothetical relationship and construct the explanatory model of "rural soundscape perception → tourism nostalgia → place attachment → environmental restoration perception". In terms of data processing, this study used SPSS23.0 software to complete all descriptive statistical analysis of data and data reliability and validity tests, AMOS24.0 software was used to conduct Confirmatory Factor Analysis (CFA), fit measurement of the model, and path coefficient analysis, followed by the application of Bootstrap methodology for the mediation effect test of the path. Finally, the interaction effect model was used to analyze the moderating effect.

## 4. Results

### 4.1. Reliability and validity tests of the scale

Firstly, using SPSS23.0 software, Cronbach's alpha coefficients of the variables involved in the conceptual model were calculated separately, and the results showed that Cronbach's alpha coefficients of each latent variable were higher than 0.8, thus proving that the questionnaire scale had good reliability. Then the questionnaire data were subjected to CFA using AMOS 24.0 software with a maximum likelihood estimate (MLE). The data results showed that the combined reliability (CR) of all variables was higher than 0.8, thus further validating the internal consistency and stability of the variables in the questionnaire scale.

In this research, the validity of the questionnaire scales was tested in four directions, namely, content validity, criterion validity, construct validity, and conjoint validity [134]. Since criterion validity coefficients are more difficult to calculate and use, and this is generally not reported in tests [135], criterion validity tests will not be conducted in this research paper. As mentioned earlier, in the design process of the questionnaire scale items, the items of each variable are derived from the mature scales in the literature, and the items are modified to a certain extent for the special environment of rural tourism, which ensures that the questionnaire scale has better content validity.

Construct validity refers to the degree and ability of the content of a questionnaire scale to measure theoretical abstract concepts, and is usually determined by the variance contribution rate of the first principal component of each variable, which is generally required to be greater than 40% to be acceptable [134]. The results showed that the contribution rate of the first principal component of each scale was as follows: rural soundscape perception (63.72%), personal nostalgia (72.87%), historical nostalgia (75.05%), place dependence (80.69%), place identity (80.19%), charisma (78.92%), ductility (77.13%), compatibility (75.00%) and distance (77.19%), all of which are greater than 40%. This shows that the questionnaire scale items have a greater contribution to the corresponding variables, so the construct validity of the questionnaire scale is good.

Conjoint validity is a test of the convergent and discriminant validity of a questionnaire scale [134]. Convergent validity is tested by the standardized loading coefficients of the measurement question items of each variable and its average variance extracted (AVE) [136]. From the results of the CFA in Table 3, it can be seen that the AVE of each latent variable is greater than 0.5, which indicates that the measurement items can explain most of the variance of each latent variable, and the standardized factor loadings of all the measurement items on

**Table 3. Confirmatory factor analysis.**

| Variables | Standardized factor loading | Cronbach's α coefficient | CR | AVE |
|---|---|---|---|---|
| Rural soundscape perception | 0.785 | 0.856 | 0.858 | 0.554 |
| | 0.792 | | | |
| | 0.786 | | | |
| | 0.797 | | | |
| | 0.524 | | | |
| Personal nostalgia | 0.739 | 0.938 | 0.937 | 0.682 |
| | 0.783 | | | |
| | 0.798 | | | |
| | 0.836 | | | |
| | 0.881 | | | |
| | 0.864 | | | |
| | 0.871 | | | |
| Historical nostalgia | 0.739 | 0.932 | 0.923 | 0.669 |
| | 0.783 | | | |
| | 0.798 | | | |
| | 0.836 | | | |
| | 0.881 | | | |
| | 0.864 | | | |
| Place dependence | 0.836 | 0.940 | 0.939 | 0.756 |
| | 0.851 | | | |
| | 0.901 | | | |
| | 0.872 | | | |
| | 0.888 | | | |
| Place identity | 0.851 | 0.923 | 0.923 | 0.751 |
| | 0.831 | | | |
| | 0.914 | | | |
| | 0.870 | | | |
| charisma | 0.857 | 0.909 | 0.911 | 0.720 |
| | 0.841 | | | |
| | 0.859 | | | |
| | 0.838 | | | |
| ductility | 0.814 | 0.898 | 0.901 | 0.696 |
| | 0.811 | | | |
| | 0.851 | | | |
| | 0.860 | | | |
| compatibility | 0.834 | 0.915 | 0.917 | 0.688 |
| | 0.838 | | | |
| | 0.802 | | | |
| | 0.818 | | | |
| | 0.856 | | | |
| distance | 0.840 | 0.925 | 0.927 | 0.720 |
| | 0.762 | | | |
| | 0.889 | | | |
| | 0.871 | | | |
| | 0.877 | | | |

**Table 4. Discriminant validity analysis.**

| Variables | averages | standard deviation | The square root of AVE | | | | | | | | |
|---|---|---|---|---|---|---|---|---|---|---|---|
| Rural soundscape perception | 4.16 | 0.649 | 0.744 | | | | | | | | |
| Personal nostalgia | 4.23 | 0.688 | 0.751** | 0.825 | | | | | | | |
| Historical nostalgia | 4.08 | 0.768 | 0.719** | 0.782** | 0.817 | | | | | | |
| Place dependence | 3.93 | 0.854 | 0.602** | 0.660** | 0.672** | 0.869 | | | | | |
| Place identity | 4.00 | 0.805 | 0.607** | 0.657** | 0.660** | 0.863** | 0.866 | | | | |
| charisma | 4.12 | 0.725 | 0.659** | 0.674** | 0.687** | 0.773** | 0.806** | 0.848 | | | |
| ductility | 4.13 | 0.717 | 0.649** | 0.672** | 0.679** | 0.771** | 0.808** | 0.833** | 0.834 | | |
| compatibility | 4.10 | 0.727 | 0.674** | 0.690** | 0.726** | 0.767** | 0.774** | 0.862** | 0.835** | 0.829 | |
| distance | 4.20 | 0.722 | 0.616** | 0.635** | 0.659** | 0.721** | 0.737** | 0.801** | 0.833** | 0.792** | 0.848 |

* $p < 0.05$

** $p<0.01$, Diagonal is the square root of AVE.

their corresponding variables are greater than 0.5, and all of them are highly significant at 5% statistical level, which indicates that the questionnaire scales have good convergent validity.

Discriminant validity is demonstrated by comparing the square root of AVE of latent variables with the absolute value of the correlation coefficient between the variables, if the former is greater than the latter, it indicates that there is good discriminant validity between the variables [137]. The results of the statistical analysis of AMOS24.0 software are shown in Table 4. The square root of AVE of each latent variable is greater than the Pearson correlation coefficient between the latent variable and other variables, and only the square root of AVE of rural soundscape perception is 0.744, which is slightly smaller than that of personal nostalgia (0.751), the difference is relatively small, so it can be assumed that there is a discriminant validity between the scales. Therefore, overall the scales designed in this study have good discriminant validity.

## 4.2. Structural equation modeling test

In this research, the fitness of the structural equation model was mainly tested using the commonly used fitness metrics shown in Table 5. Mulaik et al. (1989) [138] stated that when the sample size is greater than 500, the specified value of $\chi2/df$ is less than 5 instead of the usual value of 3. Wu (2010) [139] argued that the loosely specified value of $\chi2/df$ is 5. Given this, the model's value of $\chi2/df$ is 3.196 is within the specified acceptable range. The NFI value of 0.876 is above 0.8, though it falls short of the desired value of greater than 0.9. Therefore, the fit is acceptable. The RMSEA value of 0.066 meets the requirement of being less than 0.08. The TLI and CFI values were 0.904 and 0.911, respectively, which were greater than 0.9. The PNFI and PGFI values are 0.811 and 0.699, respectively, both of which are greater than 0.5. Therefore, as per the comparison of various model fit indicators, the model in this study has a good fit.

**Table 5. Evaluation of model fitness indicators.**

| Indicators | Absolute Fit Indicator | | Value-Added Fitness Indicator | | | Simplicity Fitness Indicator | |
|---|---|---|---|---|---|---|---|
| Specific indicators | $\chi2/df$ | RMSEA | NFI | TLI | CFI | PNFI | PGFI |
| Judgment Criteria | (1–5) | <0.08 | >0.9 | >0.9 | >0.9 | >0.5 | >0.5 |
| Measurement results | 3.196 | 0.066 | 0.876 | 0.904 | 0.911 | 0.811 | 0.699 |
| Fitness Evaluation | Ideal | Ideal | Acceptable | Ideal | Ideal | Ideal | Ideal |

**Table 6. Evaluation of model fitness indicators.**

| Hypothesis | pathway relationship | Unstandardized factor loading | Standardized factor loading | SE | Z | p |
|---|---|---|---|---|---|---|
| H1 | Rural soundscape Perception → Charisma | 0.174 | 0.164 | 0.049 | 3.54 | *** |
| | Rural soundscape perception → Compatibility | 0.152 | 0.145 | 0.049 | 3.10 | ** |
| | Rural soundscape perception→ Ductility | 0.176 | 0.165 | 0.05 | 3.53 | *** |
| | Rural soundscape perception→ Distance | 0.144 | 0.136 | 0.057 | 2.54 | * |
| H2 | Rural soundscape Perception→ Personal nostalgia | 0.797 | 0.751 | 0.031 | 25.60 | *** |
| | Rural soundscape Perception→ Historical nostalgia | 0.851 | 0.719 | 0.037 | 23.25 | *** |
| H3 | Personal nostalgia → Charisma | 0.056 | 0.056 | 0.040 | 1.39 | 0.163 |
| | Personal nostalgia → Compatibility | 0.071 | 0.072 | 0.040 | 1.78 | 0.075 |
| | Personal nostalgia → Ductility | 0.052 | 0.052 | 0.041 | 1.27 | 0.203 |
| | Personal nostalgia → Distance | 0.055 | 0.055 | 0.047 | 1.18 | 0.238 |
| | Historical nostalgia → Charisma | 0.113 | 0.126 | 0.036 | 3.19 | ** |
| | Historical nostalgia → Compatibility | 0.098 | 0.111 | 0.035 | 2.77 | ** |
| | Historical nostalgia→ Ductility | 0.208 | 0.23 | 0.036 | 5.77 | *** |
| | Historical nostalgia → Distance | 0.151 | 0.168 | 0.041 | 3.68 | *** |
| H5 | Rural soundscape perception → Place dependence | 0.181 | 0.141 | 0.075 | 2.39 | * |
| | Rural soundscape perception →Place identity | 0.202 | 0.167 | 0.072 | 2.82 | ** |
| H6 | Place dependence → Charisma | 0.158 | 0.191 | 0.029 | 5.54 | *** |
| | Place dependence → Compatibility | 0.151 | 0.184 | 0.028 | 5.29 | *** |
| | Place dependence → Ductility | 0.201 | 0.241 | 0.029 | 6.93 | *** |
| | Place dependence → Distance | 0.178 | 0.215 | 0.033 | 5.39 | *** |
| | Place identity → Charisma | 0.392 | 0.448 | 0.030 | 13.06 | *** |
| | Place identity → Compatibility | 0.405 | 0.468 | 0.030 | 13.55 | *** |
| | Place identity → Ductility | 0.269 | 0.305 | 0.030 | 8.82 | *** |
| | Place identity → Distance | 0.302 | 0.344 | 0.035 | 8.70 | *** |
| H8 | Personal nostalgia → Place dependence | 0.344 | 0.285 | 0.059 | 5.86 | *** |
| | Personal nostalgia → Place identity | 0.329 | 0.288 | 0.056 | 5.89 | *** |
| | Historical nostalgia → Place dependence | 0.396 | 0.365 | 0.05 | 7.92 | *** |
| | Historical nostalgia → Place identity | 0.338 | 0.33 | 0.048 | 7.11 | *** |

* p < 0.05

** p<0.01

*** p<0.001, "→" indicates a path influence relationship

Meanwhile, structural equation modeling was used to further explore the path relationship between rural soundscape perception, tourism nostalgia, place attachment, and environmental restoration perception. And the model results are formed in Fig 4 according to the model validation results (see Table 6) it is shown:

## 4.3. Mediation analysis

The research hypothesis suggests three paths leading from rural soundscape perception to environmental restoration perception. The first mediating path (H4): rural soundscape perception → tourism nostalgia → environmental restoration perception. The 2nd mediating path (H7): rural soundscape perception → place attachment → environmental restoration perception. The 3rd mediating path (H9): rural soundscape perception → tourism nostalgia → place attachment → environmental restoration perception. To investigate the mediating role of tourism nostalgia and place attachment between rural soundscape perception and perceptions of environmental restoration, the research adopted Hayes et al.'s (2009) [140]

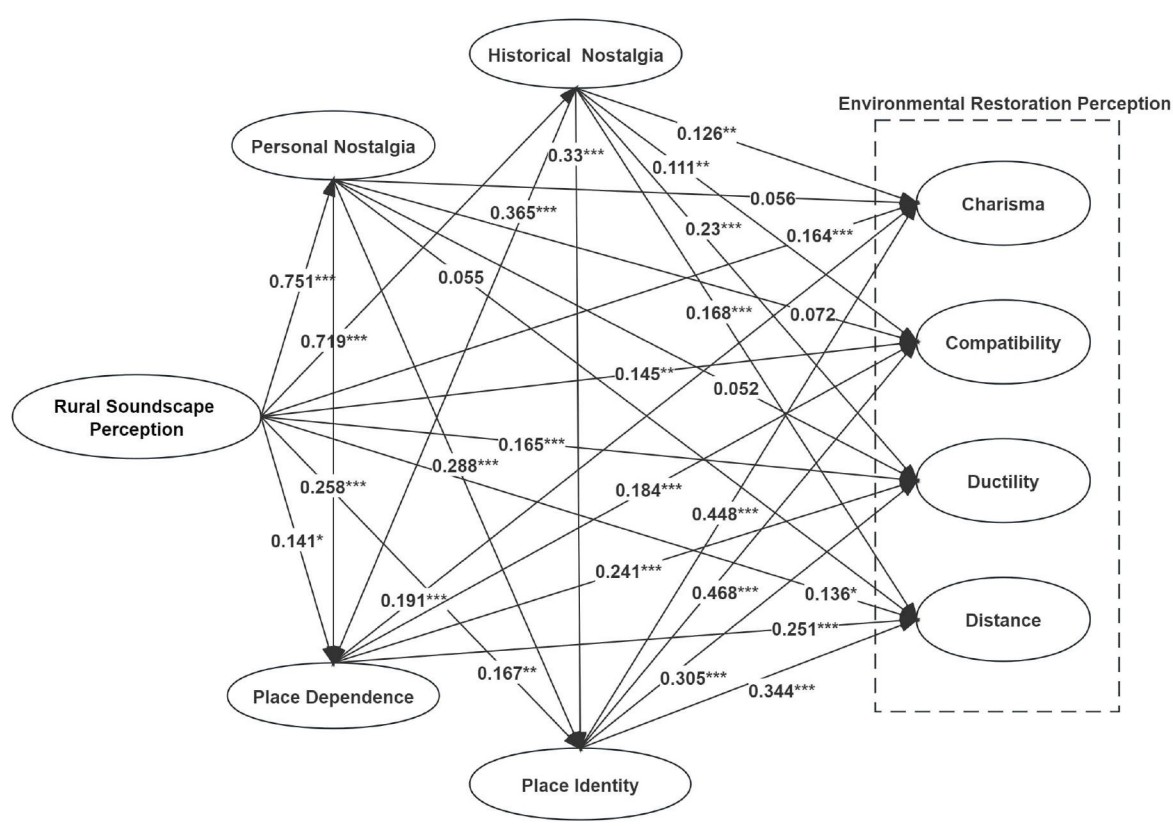

**Fig 4. Model results graph.**

suggestion of setting the Bootstrap sampling number to 2000 and conducting chained mediation effect analysis with a 95% confidence level interval.

As can be seen from Table 7, the upper and lower bounds of Bootstrap 95% CI for hypothesized paths H4, H7, and H9 are [0.089~0.241], [0.029~0.161] and [0.224~0.350], respectively, which do not include 0. Therefore, the mediating paths are all significant and the hypotheses are valid. Among them, the mediating effect value of H9 is the largest, i.e., the chain mediating effect of tourism nostalgia and place attachment is the strongest. This was followed by H4 with a mediator effect value of 0.170. H7 had the smallest effect value, i.e., the mediator effect value for place attachment was the weakest. Moreover, the chain-mediated effect of H9 is higher

**Table 7. Mediation analysis.**

|  | Effect value | Boot standard error | Boot LLCI | Boot ULCI | Relative Effect Percentage |
|---|---|---|---|---|---|
| Total effect | 0.724 | 0.033 | 0.658 | 0.788 | 100% |
| direct effect | 0.160 | 0.034 | 0.092 | 0.227 | 22.1% |
| H4 | 0.170 | 0.039 | 0.089 | 0.241 | 23.5% |
| H7 | 0.097 | 0.034 | 0.029 | 0.161 | 13.4% |
| H9 | 0.297 | 0.032 | 0.224 | 0.35 | 41% |

H4 'Rural soundscape perception → tourism nostalgia → environmental restoration perception'

H7 'rural soundscape perception → place attachment → environmental restoration perception'

H9 'rural soundscape perception → tourism nostalgia → place attachment → environmental restoration perception'.

**Table 8. Moderation analysis.**

| Path | standard error | t | p | β |
|---|---|---|---|---|
| Rural soundscape perception → Environmental restoration perception | 0.034 | 20.905 | 0.000** | 0.689 |
| Number of trips → Environmental restoration perception | 0.029 | 0.009 | 0.993 | 0 |
| Interaction items → Environmental restoration perception | 0.039 | -1.071 | 0.285 | -0.035 |

than the direct effect of ' rural soundscape perception → environmental restoration perception.' Therefore, this research suggests that the mechanism by which rural soundscape perception affects the restorative effect is mainly transmitted through multiple-chain-mediated paths.

## 4.4. Moderation analysis

Regarding the test of moderating effects, the latent variable interaction effect structural equation modeling method with the unconstrained method proposed by Wen et al. (2010) [141] has been widely cited in academic papers and journals both at home and abroad due to its simplicity, accuracy, lack of error and good robustness of results. Based on this, this research adopts this method to test the moderating effect of the number of rural tourism experiences.

As can be seen in Table 8, the objective of the path "rural soundscape perception → environmental restoration perception" is to investigate the effect of the independent variable (rural soundscape perception) on the dependent variable (environmental restoration perception) without considering the interference of the moderator variable (number of trips). Table 7 shows that the independent variable (rural soundscape perception) presents significance (t = 21.801, p = 0.000<0.001), implying that rural soundscape perception will have a significant influence relationship on Environmental restoration perception, which again verifies that H1 is valid from another perspective. In addition, the path of the "interaction term → environmental restoration perception" shows that the interaction between rural soundscape perception and the number of trips does not have any significance (t = -1.071, p = 0.285>0.05). Hence, the number of trips taken to rural areas does not moderate the relationship between rural soundscape perception and environmental restoration perception. Moreover, when considering the "interaction term → environmental restoration perception" pathway, it is evident that the interaction term linking rural soundscape perception and the number of trips is not significant (t = -1.071, p = 0.285>0.05). This implies that the number of trips to rural areas does not moderate the structural relationship between rural soundscape perception and environmental restoration perception. Therefore, H10 is not valid.

## 5. Discussion

In the background of high urbanization, rural tourism has become an ideal choice for urban residents to temporarily escape from the fast-paced life in the city and meet their psychological needs to return to freedom. Therefore, paying attention to rural soundscape is of great significance to the development of rural tourism research. Based on the restorative environment theory, this research aims to determine the applicability of each research scale through CFA, and then explore the relationship between rural soundscape perception and environmental restorative perception through SEM path analysis. The results showed that each scale had good reliability and validity. In addition, 10 research paths were established in the model of this study, of which 8 research paths were established. Additionally, the mediating role of tourism nostalgia and place attachment between rural soundscape perception and environmental restoration perception was identified. The following is a discussion and analysis of the research model paths. Additionally, the mediating role of tourism nostalgia and place attachment between

rural soundscape perception and environmental restoration perception was identified. The following is a discussion and analysis of the research model paths.

It was found that Rural Soundscape Perception significantly and positively affects Environmental Restoration Perception, which indicates that when people perceive Rural Soundscape Perception during rural tourism, fatigue and stress can be restored, and an individual's Environmental Restoration Perception will increase. This finding is consistent with the findings of scholars such as Lee (2018) [142] and Sang (2020) [59] that rural soundscapes can have a restorative effect on people's psychology. Most of the previous studies on the restorative nature of rural soundscapes were conducted by selecting Western villages or discussing the restorative nature of soundscapes in Chinese villages by viewing pictures and playing audio recordings [143–145], e.g., Ren et al. [143] explored the restorative nature of rural soundscapes in China in a simulation room by using pictures of Chinese villages in the landscapes and audio recordings and audio recordings. However, there are differences between actual and simulated environments, and Thorogood et al. (2016) [146] argued that laboratory assessments cannot fully express how subjects' stress and comfort are affected by soundscapes. Meanwhile, rural soundscapes around the world can vary slightly depending on the environment and cultural life. Chinese countryside soundscape is characterized by poetry and natural simplicity [147], and Zhang Chao, a Chinese literati in the Qing Dynasty, described Chinese countryside soundscape in The Shadow of a Phantom: "Listening to the sound of birds in the spring, cicadas in the summer, insects in the autumn, snow in the winter, chess in the daytime, xiao under the moonlight, the wind of the pines in the mountains, and the sound of hei nai between the waters", and this sound environment is considered a typical Chinese rural soundscape [148]. In addition, the sounds of chickens and dogs, rain beating on bananas, vendors yelling, cowherds herding cattle, and labor trumpets are all unique to the Chinese countryside, so the simulation of Chinese countryside environments through indoor experiments may result in a lack of authenticity of the Chinese countryside, thus making it difficult to truly explore the restorative effects of the environment of the Chinese countryside soundscape; therefore, this paper further supports Ren's research based on the findings of the field research in the Chinese countryside. Therefore, this paper further supports the findings of Ren and other scholars based on field research in the Chinese countryside. In addition, this paper also provides empirical arguments for previous scholars' conclusions about the restorative effects of soundscapes [149,150]. At the same time, few studies have focused on the association between Rural Soundscape Perception and Environmental Restoration Perception in the context of tourism, and the findings of this study may fill this research gap.

From the previous section, it can be seen that there are three mediating paths in this study, and the mediating effects of all three paths are significant, among which the chain mediating effect of "rural soundscape perception→tourism nostalgia→place attachment→environmental restoration perception" has the strongest value. This was followed by the mediating effect value of "rural soundscape perception → place attachment → environmental restoration perception", and the mediating effect value of "rural soundscape perception → place attachment → environmental restoration perception" was the weakest. In addition, the chain mediation effect is stronger than the direct effect of "rural soundscape perception → environmental restoration perception". Therefore, this study suggests that the mechanism of rural soundscape perception on environmental restoration perception is mainly transmitted through the chained multiple mediation path. The chained multiple mediation paths are specifically manifested in the fact that tourists perceive rural soundscapes rich in history, culture, regional geography, and other attributes through sensation and hearing in the course of rural tourism, thus evoking subjective experiences and historical memories [151], which give rise to nostalgia. Nostalgia and place are closely intertwined and connected [152]. They combine temporal memory and

emotional experience creatively [153], evoking people's emotions of being attached to a certain place. Place attachment is closely linked to the generation of restorative feelings [154], and perceiving environmental restoration is produced with the emotion of place attachment. This research provides further empirical evidence for the conclusion above and reveals a complex psychological path of "rural sound perception → nostalgic tourism → place attachment → environmental restoration perception," which includes two emotions of nostalgic tourism and place attachment. Recognizing this psychological path provides certain theoretical guidance for explaining the restorative effect of rural soundscapes.

This study found that rural soundscape perception has a significant positive effect on tourism nostalgia. This finding validates the assertion of Payne (2013) [57] and Hall et al. (2013) [151] that soundscapes can trigger memories of past times. In addition, the research hypothesis H3 that tourism nostalgia has a significant positive effect on Environmental Restoration Perception is partially valid, where historical nostalgia significantly and positively affects Environmental Restoration Perception, while personal nostalgia does not have a significant effect on Environmental Restoration Perception. This result can be explained by several reasons. Firstly, This research reflects on the definition of tourism nostalgia: personal nostalgia consists mainly of events actually experienced by individuals in the past [126]. Some researchers have argued that nostalgia is a negative emotion involving sadness and pain in an individual's past. It is the process of people's remembrance of the good things in the past, and when the good things that have passed away no longer exist or cannot be reproduced, people feel sadness and helplessness [155], and this nostalgic emotion of feeling deep remembrance for the things that have passed away is more likely to be the motivation for individuals to escape from the daily routine and to seek after the other, and it becomes an effective way for people to escape from modernity [66]. Therefore, personal nostalgia may be more closely related to tourism nostalgia and have a less significant effect on Environmental Restoration Perception, which has been confirmed by previous scholars' studies [126,156]. Historical nostalgia focuses on the scenes of people's lives in the past, which can be events that happened in history or memories of a specific historical era [126]. Nostalgia has an aesthetic character, and the subject of nostalgia tends to desire and pursue beautiful things [66]. Since historical nostalgia is an individual's imagination of the past that he or she has not experienced, there is a large space for imagination, therefore, people tend to miss the beautiful and happy historical memories, which is a kind of positive nostalgia emotion that helps to alleviate people's mental stress and physical and mental exhaustion, and then promotes restorative nature. Secondly, thinking from the perspective of social phenomenon: the content of personal nostalgia requires real experience, so it is highly subjective, i.e., different experiences of individuals lead to different nostalgia produced by individuals [126]. With the acceleration of urbanization, more and more people are moving from the countryside to the cities, and China's urbanization rate has reached 64.72% as of 2021. The accelerated urbanization, on the one hand, makes many people unfamiliar with the countryside and its soundscape, especially contemporary children or young people, because most of them grew up in the city and know more about the countryside from electronic devices and books, and thus may have a focused and biased memory of the countryside [157]. Therefore, then, the lack of authentic countryside experiences leads to unfamiliarity with the countryside, which in turn makes it difficult for tourists to develop personal nostalgia during tourism. It has been confirmed that the more familiar an individual is with the environment he or she is in, the stronger his or her Environmental Restoration Perception is [106]. So when the familiarity of a place is not enough for tourists to produce personal nostalgia, the lower their Environmental Restoration Perception, so the effect of personal nostalgia on Environmental Restoration Perception is not significant. On the other hand, the acceleration of urbanization has also changed the appearance of the traditional countryside, making the artificial and

mechanical sounds in the countryside increase [158], which makes the tranquil countryside soundscape different from the countryside soundscape in people's recollections, which leads to feelings of sadness and helplessness, and this negative emotion is not conducive to promoting restorativeness [159]. Currently, there is less research on the relationship between tourism nostalgia and environmental restoration perception, and the results of this study provide an important addition and refinement to research in this area.

This research confirms that rural soundscape perception has a significant positive effect on place attachment, which is consistent with Fang et al.'s (2022) [103] finding that rural sound as a perceptual resource of rural tourism positively affects place attachment. It can be seen that tourists' rural soundscape perception deepens the connection between people and place. Both dimensions of place attachment significantly and positively affect environmental restoration perception, which suggests that tourists' place attachment feelings promote their environmental restoration perception. Previous studies have explored the relationship between place attachment and environmental restoration perception based on contexts such as native places [106] and urban parks [107,108], and the findings are consistent with the present research: there is a positive association between place attachment and environmental restoration perception in the findings. However, no scholars have yet focused on the association between tourists' place attachment and environmental restoration perception in rural soundscape contexts, and this study enriches the research in this area.

This study found that tourism nostalgia in the context of rural soundscapes has a significant positive effect on place attachment, and this conclusion is similar to previous studies [115]. For tourists, nostalgic emotion has a positive effect on place attachment [160], the stronger their nostalgic emotion, the stronger their attachment to the tourist place [161], and the soundscape will stimulate nostalgia [162,163]. In other words, tourists in the context of rural soundscapes produce stronger nostalgic emotions, which trigger place attachment. Therefore, nostalgic emotions help promote place attachment. This study further investigated the association between tourists' tourist nostalgia and place attachment in the context of rural soundscapes.

The study indicates that the number of trips does not moderate the relationship between perceived rural soundscape and environmental restoration perception. It is suggested that, in the context of rural soundscape perception, the number of trips does not influence environmental restoration perception. This finding is similar to Guo et al.'s (2014) [164] study, which found that the number of trips does not have a significant impact on tourists' perception of environmental restoration. It is also important to note that not all soundscapes in rural environments are comforting sounds, and certain rural tourist locations may be filled with a large amount of uncomfortable noise, such as mechanical sounds. Therefore, it is the environment in which the traveler is located that should influence individual restorative effects, not the number of trips, and thus the number of rural travel experiences may not play a moderating role in the structural relationship between Rural Soundscape Perception and Environmental Restoration Perception. It has been found that the length of travel time has a significant effect on the effect of restorative environmental perception, and the longer the stay, the higher the restorative experience [165], as well as the factors of place memory, place dependence and place identity, and familiarity of the environment positively affect people's Environmental Restoration Perception [166,167], so it can be seen that tourists' Environmental Restoration Perception may be more related to the length of tourists' travel time, familiarity of the environment, local memory and other factors, while it is not closely related to the number of tours.

## 6. Conclusions

This study explores the environmental restorative effect of tourists based on the perspective of soundscape perception and introduces the relationship of a total of four variables, namely,

rural soundscape perception, tourism nostalgia, place attachment, and environmental restoration perception, to rural tourist places for empirical research, and constructs a mechanism for the influence of rural soundscape perception on environmental restoration perception. At the same time, it also reveals the complex path of "rural soundscape perception→tourism nostalgia→place attachment→environmental restoration perception", which provides a new perspective for the understanding of the mechanism of the rural environment and people's health, and to a certain extent, broadens the research of the combination of soundscape ecology, landscape design, tourism, and environmental psychology. Although research on the restorative effects of soundscapes has been widely used in previous restorative environmental research, there have been fewer studies on the mechanisms by which soundscapes influence the restorative effects of the environment, and insufficient emphasis has been placed on the restorative aspects of rural soundscapes. Based on the results of the study, the main conclusions are as follows: (1) Rural soundscape perception had a significant positive effect on tourism nostalgia, place attachment, and environmental restoration perception. (2) Tourism nostalgia and place attachment mediated the relationship between rural soundscape perception and environmental restoration perception. (3) The number of trips did not play a moderating role in the structural relationship between rural soundscape perception and environmental restoration perception. In addition, it is clear from the results that satisfying tourists' rural soundscape perceptions can stimulate tourism nostalgia and place attachment, thus giving them environmental restoration perception that helps them relieve stress and fatigue during tourism.

## 7. Implications and limitations

### 7.1. Implications

The findings of this paper show that the soundscape environment of rural tourist sites has a positive impact on tourists' restorative properties. Therefore, scenic area managers can enhance the environmental restorative effect of tourists as well as the quality of tourism experience through the optimization of scenic soundscape design. Based on this, the following suggestions for optimizing the soundscape environment of rural tourism destinations are proposed for reference and reference.

(1) Focus on tourists' soundscape expectations and design positive factors. The rural soundscape environment is a comprehensive environment that contains natural soundscape and rural cultural soundscape. The natural soundscape includes the sounds of birdsong, running water and wind blowing leaves around the village, which help people relax and feel the rural life. The rural cultural soundscape includes traditional folk music, instrumental music, dance performances and local dialects, which let people feel the charm of traditional culture and enhance the sense of cultural identity. In the rural soundscape design, we need to study and understand the characteristics of the natural soundscape and rural cultural soundscape, and reasonably integrate them into the design to provide a better experience for the villagers and tourists, so that the soundscape design can enhance the natural atmosphere and cultural heritage of the rural environment, so that people can better feel the joy of rural life, and thus can promote the environmental restorative effect.

(2) Optimize the configuration of audiovisual elements and control negative factors. Rural tourists favor quiet, tranquil atmosphere and characteristic activities, but due to large-scale tourism development and tourist crowding, traditional villages become noisy. Valuable soundscape resources such as natural sounds, sounds of folk activities and farming sounds have not been fully utilized. Therefore, the government should plan and manage beautiful soundscapes in the rural environment. While using positive factors to shape the traditional village environment, it should also try to avoid negative factors. For example, unnecessary and

incongruous sounds can be controlled and reduced by planting hedges and installing sound barriers that can shield road traffic noise and mechanical noise. At the same time, some natural sounds can be created to mask the noise.

(3) Protect the traditional soundscape of the countryside and inherit the cultural factors. Tourists have a high preference for rural natural soundscape and rural cultural soundscape, and rural soundscape perception has a positive effect on environmental restorative perception. Therefore, increasing and highlighting the rural natural soundscape and cultural soundscape can influence the restorative perception of tourists to a certain extent and improve their traveling experience. Scenic area managers should first carry out natural ecological environment protection, so as to protect the rich and diverse natural soundscape in the rural scenic area. Secondly, the soundscape with rural cultural characteristics in the scenic area should be publicized and protected, such as: playing natural pure music, local representative music, etc. to reduce the noise impact inside the scenic area; in addition, through stage performances, film and television creations and other ways, the soundscape with local rural characteristics will be disseminated and diffused, so as to achieve the effect of inheritance and protection.

## 7.2. Limitations

There are still some shortcomings in this study: first, in the design of the rural soundscape perception questionnaire scale, due to the lack of corresponding perception scales in the current academic world, the rural soundscape perception scale used in this study is a scale developed and designed independently based on reference to other literature. Although the scale meets the validity and reliability requirements, there may still be issues such as the insufficient evaluation dimensions and biased selection of indicators, which need to be optimized in the future to better guide the study of rural soundscape perception. Second, this research utilizes the RPRS scale to measure the restorative nature of the environment, which is mainly related to the part of environmental perception, and there is no actual measurement of the restorative environment on the human body's physiological effects such as blood pressure and heart rate, etc. In the future, this research can measure corresponding physiological indexes, such as myoelectric value, blood pressure, brain wave, etc., to further reveal the medical value of rural soundscape perception on the human body's health and its functioning mechanism.

## Supporting information

**S1 File. Minimal data set.**
(XLSX)

## Acknowledgments

The authors are grateful to all the investigators and respondents who participated in this study for their contribution to data collection.

## Author Contributions

**Conceptualization:** ShuangQuan Zhang.

**Data curation:** Hui Yang.

**Formal analysis:** Hui Yang.

**Funding acquisition:** ShuangQuan Zhang.

**Investigation:** Hui Yang.

**Methodology:** Hui Yang, ShuangQuan Zhang.

**Software:** Hui Yang.

**Validation:** Hui Yang.

**Visualization:** Hui Yang.

**Writing – original draft:** Hui Yang.

**Writing – review & editing:** ShuangQuan Zhang.

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
