## [Decision Letter · Decision Letter 0]

16 Oct 2023

PONE-D-23-26044A study on the mechanism of the impact of rural soundscape perception on environmental restoration: An empirical study based on the Taohuayuan Scenic Area in Changde, ChinaPLOS ONE

Dear Dr. Zhang,

Thank you for submitting your manuscript to PLOS ONE. After careful consideration, we feel that it has merit but does not fully meet PLOS ONE’s publication criteria as it currently stands. Therefore, we invite you to submit a revised version of the manuscript that addresses the points raised during the review process.Both reviewers raised some flaws in the manuscript. I recommend to revise it carefully in an effort to address their comments and to improve the paper.

We look forward to receiving your revised manuscript.

Kind regards,

Federica Biassoni

Academic Editor

PLOS ONE

Journal Requirements:

2. You indicated that ethical approval was not necessary for your study. We understand that the framework for ethical oversight requirements for studies of this type may differ depending on the setting and we would appreciate some further clarification regarding your research. Could you please provide further details on why your study is exempt from the need for approval and confirmation from your institutional review board or research ethics committee (e.g., in the form of a letter or email correspondence) that ethics review was not necessary for this study? Please include a copy of the correspondence as an ""Other"" file.

5. We note that Figure 2 in your submission contain map images which may be copyrighted. All PLOS content is published under the Creative Commons Attribution License (CC BY 4.0), which means that the manuscript, images, and Supporting Information files will be freely available online, and any third party is permitted to access, download, copy, distribute, and use these materials in any way, even commercially, with proper attribution. For these reasons, we cannot publish previously copyrighted maps or satellite images created using proprietary data, such as Google software (Google Maps, Street View, and Earth). For more information, see our copyright guidelines: http://journals.plos.org/plosone/s/licenses-and-copyright.

1.) You may seek permission from the original copyright holder of Figure 2 to publish the content specifically under the CC BY 4.0 license.  

2.) If you are unable to obtain permission from the original copyright holder to publish these figures under the CC BY 4.0 license or if the copyright holder’s requirements are incompatible with the CC BY 4.0 license, please either i) remove the figure or ii) supply a replacement figure that complies with the CC BY 4.0 license. Please check copyright information on all replacement figures and update the figure caption with source information. If applicable, please specify in the figure caption text when a figure is similar but not identical to the original image and is therefore for illustrative purposes only.

6. Please ensure that you refer to Figure 2 to 4 in your text as, if accepted, production will need this reference to link the reader to the figure.

7. We note you have included a table to which you do not refer in the text of your manuscript. Please ensure that you refer to Table 1 in your text; if accepted, production will need this reference to link the reader to the Table.

Reviewers' comments:

Reviewer's Responses to Questions

**Comments to the Author**

1. Is the manuscript technically sound, and do the data support the conclusions?

Reviewer #1: Yes

Reviewer #2: Yes

2. Has the statistical analysis been performed appropriately and rigorously? 

Reviewer #1: Yes

Reviewer #2: Yes

3. Have the authors made all data underlying the findings in their manuscript fully available?

Reviewer #1: Yes

Reviewer #2: Yes

4. Is the manuscript presented in an intelligible fashion and written in standard English?

Reviewer #1: Yes

Reviewer #2: No

5. Review Comments to the Author

Reviewer #1: This study contains interesting ideas however improvements are required:

Abstract – please include who are the respondents of the study.

Introduction - this section should contain the knowledge gap, and to explain why it is important to address the gaps in knowledge.

The literature review requires updating as majority of them are old.

In the discussion section, the authors could add further explanation on the insignificant moderation relationship.

The conclusion section should include discussion that are connected to the empirical results to the theoretical as well as literature.

Implication section should include practical/managerial implications as well as recommendations to the stakeholders/managers/tourism agencies.

Reviewer #2: Recommendation: Major Revision

Manuscript Number: PONE-D-23-26044

Title: A study on the mechanism of the impact of rural soundscape perception on environmental restoration: An empirical study based on the Taohuayuan Scenic Area in Changde, China

1. Overview and general recommendation

The idea of a framework to research on soundscape and restoration. However, I think that the descriptions of some very important points were inadequate. I recommend that a major revision is warranted. I explain my concerns in more detail below. I ask that the authors specifically address each of my comments in their response.

2. Major comments

1) It is suggested to add key words in the manuscript.

2) It is suggested to improve the framework, especially the background, literature review, methods and materials.

3) The writing is not concise enough, and the content is slightly lengthy, while it dilutes the core content. It is suggested to simplify the manuscript.

4) It is suggested to further clarify the research questions.

5) In the Abstract, there is no logical connection between the first and the second sentence.

6) From Line 44 to Line 47, I don’t think “rural soundscape perception” is a “creative concept” in this manuscript. There are some researchers who have studied rural soundscape, in addition, the concept of soundscape has been proposed in the ISO standard. This paper is not conceptually innovative. The research content and research scope are also under the scope of soundscape. It is suggested to modify.

7) From Line 74 to Line 75, what is the mean of “environmental restorative effects” referred to? In the introduction, it is not explicitly illustrated.

8) From Line 102 to Line 116, it is recommended that the research questions would be clearly presented in this paragraph, and generally speaking, there are usually three research questions.

9) In terms of “Literature Review”, first of all, it is suggested a combination of literature review and introduction. Secondly, the topic of the literature review is closely related to the research questions. Thirdly, texts which are not closely related to the topic can be deleted.

10) In paragraph of 2.3 Environmental Restoration Perception, it is suggested to clarify the aim of this paragraph, and separated method description from literature review. Paragraph of 2.4 and 2.5 should be also changed in this way.

11) From Line 245 to Line 246, four dimensions of “away, extensibility, charisma, and compatibility,” are inconsistent with the words in the following methods and tables.

12) In paragraph of 3.1 Rural Soundscape Perception and Environmental Restoration Perception, there were two topics as restoration effects of rural soundscape and differences from Chinese rural soundscape and other research. It is suggested to focus on one topic in one paragraph.

13) From Line 343 to Line 348, and from Line 386 to Line 392, it is suggested to use the literatures as the theoretical supports for research hypothesis.

14) From Line 376 to Line 383, there is little explanation or theoretical hypothesis for the relationships between the elements in paragraph of 3.2, and the concepts of the words were unclear. It is suggested to modify.

15) What was the aim of paragraph 5.5 Results of Hypotheses Testing? If it is a duplicate of the previous results, it is suggested to delete it. If it is addressing a specific research question, it is recommended to describe the results.

16) Although it seems that the basic composition of the questionnaire can be implied in the paper, it is still suggested that the composition of the questionnaire be clearly presented. For example, what are the basic information of the respondents, how to ask about the rural tourism experience, and how to divide the education level.

17) In the manuscript, H3a of research hypothesis “Personal nostalgia has a significant positive effect on environmental restoration perception” was not valid. It is suggested that this point could be further discussed.

3. Minor comments

1) It is suggested to simplify the topic, such as “Impact of rural soundscape on environmental restoration: An empirical study based on the Taohuayuan Scenic Area in Changde, China”.

2) Line 204, it is suggested to delete “to fill the gap in the literature”. The manuscript focuses on the increase of measurement items in the specific study case, and whether it is universal needs to be further discussed.

3) Line 235, what is RPRS? When it first appears, it is suggested to write the full name. In addition, this scale is used in this manuscript, and it is suggested to explain the reasons to select and describe the scales in detail in methods parts.

4) In Table 1, “CH4” and “CH5” should be “CH3”and “CH4”. Please check the details.

5) Figure 2, Figure 3 and Figure 4 were not mentioned in the text. Please check the details.

6) The words do not have to capitalize the first letter, such as “convergent validity”, or “discriminant validity”. Please check the details.

7) In Table 3, it is suggested to use words phrases instead of numbers.

8) Moreover, it is suggested that the manuscript should undergo extensive English editing.

6. PLOS authors have the option to publish the peer review history of their article (what does this mean?). If published, this will include your full peer review and any attached files.

Reviewer #1: No

Reviewer #2: No

---

## [Author Response · Author response to Decision Letter 0]

16 Jan 2024

Dear Editors and Reviewers: 

We would like to sincerely thank for your help and thank for reviewers’ professional comments concerning our manuscript entitled “A study on the mechanism of the impact of rural soundscape perception on environmental restoration: An empirical study based on the Taohuayuan Scenic Area in Changde, Number: PONE-D-23-26044). Those comments are all valuable and very helpful for revising and improving our paper, as well as the important guiding significance to our research. We have revised the manuscript carefully according to the comments and suggestions of reviewers and editors and responded point by point to the comments. The revised manuscript has been edited and the revised part are highlighted in red.

Regarding the constructive suggestions made by the reviewers, they are too numerous to be shown in this box, so the details of the changes made in response to the issues raised by the reviewers will be reflected in the "Response to Reviewers". This box is primarily a point-by-point response to the valuable suggestions made by the editors.

Responds to the editor’s comments:

1.Please ensure that your manuscript meets PLOS ONE's style requirements, including those for file naming.

Thanks very much for your comments. We have revised our manuscript based on PLOS ONE's style template and hope to meet the journal's stylistic requirements.

2.You indicated that ethical approval was not necessary for your study. We understand that the framework for ethical oversight requirements for studies of this type may differ depending on the setting and we would appreciate some further clarification regarding your research. Could you please provide further details on why your study is exempt from the need for approval and confirmation from your institutional review board or research ethics committee (e.g., in the form of a letter or email correspondence) that ethics review was not necessary for this study? Please include a copy of the correspondence as an ""Other"" file.

We are grateful for the suggestion. By carefully reading the journal's requirements for ethical approval again, this study did involve Human participants, so ethical approval was applied for and relevant supporting materials were uploaded into the system.

3.In your Data Availability statement, you have not specified where the minimal data set underlying the results described in your manuscript can be found. PLOS defines a study's minimal data set as the underlying data used to reach the conclusions drawn in the manuscript and any additional data required to replicate the reported study findings in their entirety. All PLOS journals require that the minimal data set be made fully available. 

We are sorry for this problem and have uploaded our minimal data set as a Supporting Information file to the system under the file name paper data.

4.PLOS requires an ORCID iD for the corresponding author in Editorial Manager on papers submitted after December 6th, 2016. Please ensure that you have an ORCID iD and that it is validated in Editorial Manager.

In accordance with the requirements of the journal, the corresponding author of this study have made available in the Editorial Manager the ORCID iD.

5.We note that Figure 2 in your submission contain map images which may be copyrighted. All PLOS content is published under the Creative Commons Attribution License (CC BY 4.0), which means that the manuscript, images, and Supporting Information files will be freely available online, and any third party is permitted to access, download, copy, distribute, and use these materials in any way, even commercially, with proper attribution. For these reasons, we cannot publish previously copyrighted maps or satellite images created using proprietary data, such as Google software (Google Maps, Street View, and Earth). 

We really appreciate editor's valuable comment for this point. The maker of Figure 2 is one of the authors of this study and we have submitted the original copyright holder's Content Permission Form as well as written permission in the system.

6.Please ensure that you refer to Figure 2 to 4 in your text as, if accepted, production will need this reference to link the reader to the figure.

We are grateful for this suggestion. We have revised the issue and have ensured that Figures 2 to 4 are mentioned in the paper.

7.We note you have included a table to which you do not refer in the text of your manuscript. Please ensure that you refer to Table 1 in your text; if accepted, production will need this reference to link the reader to the Table.

We are grateful for this suggestion. We have revised the issue and have ensured that the paper refers to Table 1.

---

## [Decision Letter · Decision Letter 1]

27 Feb 2024

Impact of rural soundscape on environmental restoration: An empirical study based on the Taohuayuan Scenic Area in Changde, China

PONE-D-23-26044R1

Dear Dr. Zhang,

We’re pleased to inform you that your manuscript has been judged scientifically suitable for publication and will be formally accepted for publication once it meets all outstanding technical requirements.

Kind regards,

Federica Biassoni

Academic Editor

PLOS ONE

Additional Editor Comments (optional):

Reviewers' comments:

Reviewer's Responses to Questions

**Comments to the Author**

1. If the authors have adequately addressed your comments raised in a previous round of review and you feel that this manuscript is now acceptable for publication, you may indicate that here to bypass the “Comments to the Author” section, enter your conflict of interest statement in the “Confidential to Editor” section, and submit your "Accept" recommendation.

Reviewer #1: All comments have been addressed

Reviewer #2: All comments have been addressed

2. Is the manuscript technically sound, and do the data support the conclusions?

Reviewer #1: Yes

Reviewer #2: Yes

3. Has the statistical analysis been performed appropriately and rigorously? 

Reviewer #1: Yes

Reviewer #2: Yes

4. Have the authors made all data underlying the findings in their manuscript fully available?

Reviewer #1: Yes

Reviewer #2: Yes

5. Is the manuscript presented in an intelligible fashion and written in standard English?

Reviewer #1: Yes

Reviewer #2: Yes

6. Review Comments to the Author

Reviewer #1: (No Response)

Reviewer #2: The authors have revised well by reflecting the reviewers’ comments. However, there are still some minor spelling mistakes in the paper that need to be corrected. For example, in line 442, "The" should be lowercase, and in line 761, "This" should be lowercase. It is suggested to check through the whole text.

7. PLOS authors have the option to publish the peer review history of their article (what does this mean?). If published, this will include your full peer review and any attached files.

Reviewer #1: No

Reviewer #2: No

---

## [Editor Report · Acceptance letter]

7 Mar 2024

PONE-D-23-26044R1 

PLOS ONE

Dear Dr. Zhang, 

I'm pleased to inform you that your manuscript has been deemed suitable for publication in PLOS ONE. Congratulations! Your manuscript is now being handed over to our production team.

Kind regards, 

on behalf of

Dr. Federica Biassoni 

Academic Editor

PLOS ONE